# Well-Differentiated Jejunoileal Neuroendocrine Tumors and Corresponding Liver Metastases: Mesenteric Fibrogenesis and Extramural Vascular Invasion in Tumor Progression

**DOI:** 10.3390/cancers17091486

**Published:** 2025-04-28

**Authors:** Jacob M. Ranot, Jemila S. Hamid, Azita Montazeri, Kelly Harper, Christopher McCudden, Terence N. Moyana

**Affiliations:** 1Department of Pathology & Laboratory Medicine, University of Ottawa & The Ottawa Hospital, Ottawa, ON K1H 8L6, Canada; jranot@toh.ca (J.M.R.); cmccudden@eorla.ca (C.M.); 2Department of Mathematics & Statistics, University of Ottawa, Ottawa, ON K1N 6N5, Canada; jhamid@uottawa.ca; 3Department of Community Health & Epidemiology, University of Ottawa, Ottawa, ON K1N 6N5, Canada; zahra.montazeri@uottawa.ca; 4Department of Radiology: Abdominal Imaging & Intervention, University of Ottawa & The Ottawa Hospital, Ottawa, ON K1H 8L6, Canada; keharper@toh.ca

**Keywords:** jejunoileal neuroendocrine tumors, mitoses/Ki67, infiltration pattern, serotonin, mesenteric fibrogenesis, extramural vascular invasion, tumor progression

## Abstract

Patients with jejunoileal neuroendocrine tumors (JINETs) can live for many years despite liver metastases. Evidence suggests that tumor heterogeneity is prognostically important, hence the selection of Ki67 hotspots for tumor grading. According to the stepwise metastasis model, clonal hotspots should dominate the metastases. However, an alternative view holds that the polyclonality of metastases is consistent with origin from genetically heterogeneous clusters of disseminated cells. The shortcomings of Ki67 grading are also being recognized, thus renewing the search for other prognostic parameters. This study paired JINETs and hepatic metastases, analyzing various parameters. We found that JINETs and their liver metastases tend to have low proliferation rates. However, an important mechanism in the metastatic cascade appears to be mesenteric fibrogenesis. It encases vessels, which enhances extramural vascular invasion, thereby conveying clusters of tumor cells to the liver. This supports the polyclonal nature of tumor progression rather than origin from hotspots of aberrant clones.

## 1. Introduction

Gastrointestinal (GI) neuroendocrine tumors (NETs) are rare, but their incidence is increasing [1,2]. They are an interesting group of neoplasms with a relatively indolent clinical course compared to their adenocarcinoma counterparts [3]. However, a significant proportion of them have distant metastatic disease at the time of presentation, with the most common site being the liver [4,5]. Amongst GI NETs as a group, those from the jejunoileum (JI) by far account for most (>50%) hepatic metastases, and liver failure is often the cause of death [1,2]. However, it is also known that some patients can live for many years despite the metastases, but the reasons for this have not been fully elucidated [1,2,4]. Concurrently, the prevailing evidence suggests that intra- and inter-tumor NET heterogeneity may be an important determinant of outcomes [5,6,7].

Cell proliferative activity (as measured by mitoses/Ki67) has emerged as an important prognostic indicator for NE neoplasms (NENs) and is now the basis for the WHO classification [8,9]. To the extent that it can separate NETs from neuroendocrine carcinomas (NECs), it has much improved our understanding of disease biology. However, intratumoral heterogeneity due to clonal diversification can lead to discordance in grading within the primary tumor, hence the idea of selecting Ki67 hotspots [6,7,8,9,10]. By the same token, according to the stepwise metastasis model [11,12], this sets up the expectation that the constituents of the clonal hotspots would be the predominant cell population of the metastases. An alternative view holds that the polyclonal nature of metastases is consistent with origin from genetically heterogenous clusters of disseminated cells rather than clonally from single cells [13,14].

Molecular studies have shown that NETs tend to be associated with MEN1, DAXX, and ATRX mutations, and NECs tend to be associated with TP53 and RB1. However, this more commonly pertains to pancreatic NETs rather than JI NETs, which are characterized by relatively few somatic mutations and a low mutation burden [15,16,17,18]. Instead, JI NETs seem to be more associated with epigenetic dysregulation involving, e.g., DNA methylation, histone modification, or microRNAs [16,17,19]. Studies are ongoing to determine how this information can be best adopted to maximize clinical benefit [18,19,20]. It is also being recognized that there are shortcomings to using cell proliferation markers as prognostic indicators, particularly as it pertains to low-grade (G1/G2) tumors, which are more prevalent in the JI relative to the pancreas or lungs [21,22,23,24,25,26,27]. Even changing the Ki-67 cut-offs from 3% to 1% or ≥5% does not completely resolve the issue [23,24,25,26]. To this end, other recently described parameters, such as tumor infiltration pattern, mesenteric fibrogenesis, and extramural vascular invasion (EMVI), hold much promise [23,28,29,30,31,32].

In this study, we compared JI NETs and their respective liver metastases using cell proliferation indices and other parameters to assess tumor outcomes. Since patients with metastatic JI NETs can live for a long time [1,2,3,4], we extended our study as far back as 2004 in order to get a more complete picture of disease biology.

## 2. Materials and Methods

### 2.1. Case Selection

Approval for the study was obtained from our institutional Research Ethics Board. Surgical pathology records were searched for liver specimens with metastatic NENs for the 20-year period from 1 January 2004 to 31 December 2023. The search was further refined to include only cases arising from the JI. In this way, the JI primaries were paired with their corresponding hepatic metastases. The slides and tissue blocks were retrieved from archives. The JI specimens were mostly resections (Figure 1a), whereas the livers had a mixture of biopsies, wedge resections, segmental metastasectomies and lobectomies (Figure 1b). The specimens were fixed, processed, and examined according to standard procedures.

The electronic medical records and medical imaging findings (Figure 2A,B and Figure 3A,B) of the patients were reviewed to confirm the primary origin from the JI, as well as for staging. Cases where the location of the primary could not be verified (e.g., disseminated disease) were excluded. The study materials were available for most cases, except for a few where some slides and/or tissue blocks of either the JI or liver were unavailable, as shown in Table 1.

### 2.2. Histology, Special Stains, and Immunohistochemistry

The slides were reviewed to confirm the neuroendocrine nature of the tumors based on routine stains and immunohistochemistry (synaptophysin and chromogranin). They were also stained for serotonin. Tumors were graded based on mitoses/Ki67, whichever was greater. Ki67 was determined by photographing the hotspots [33]. Lymphovascular invasion (LVI) was determined based on routine stains and immunohistochemical markers (D2-40 and/or CD31). EMVI was assessed based on routine and elastin stains (Verhoeff’s van Gieson). The tumor infiltration pattern (also referred to as the advancing edge profile) was based on the overall contour of the advancing edge at the tumor/host interface [23]. When deemed necessary, additional stains were run to highlight certain features, e.g., Masson trichrome (fibrosis) and S100 (nerve twigs).

### 2.3. Statistical Analysis

Descriptive statistics characterizing patients with the JI NETs were calculated. Continuous variables were summarized using the mean and standard deviation (SD), as well as the median and inter-quartile range (IQR). The range (min, max) is also provided for all continuous variables. Categorical variables were summarized using frequencies and percentages. Association or agreement between categorical variables was quantified using percentage agreement and the kappa statistic, and statistical comparisons between relevant categorical variables were made using McNemar’s test. Our primary outcome is survival (dead = 0, alive = 1), which is a binary outcome. As such, we fitted the logistic regression model, which is a generalized linear model (GLM) with a binomial distribution and logist link [34]. We first fitted a single variable logistic regression model where the dependent variable (outcome) was survival, and each of the relevant variables was entered into the model one by one. The unadjusted odds ratio (OR) with the corresponding 95% confidence intervals (CIs) and *p*-values was then evaluated for inclusion in the final model. Next, we fitted a multivariable logistic regression model consisting of sex, age at diagnosis, MF, and tumor size. Age at follow-up was not included in the model because of multicollinearity caused by its high correlation (ρ = 0.91) with age at diagnosis. EMVI could also not be included in the model because of its high level of association (92.68% agreement) with MF. Both the unadjusted and adjusted ORs from the single variable and multivariable logistic regression models, with the corresponding 95% CI and *p*-values, are provided. The final model consists of sex, age at diagnosis, and MF, from which we calculated the survival probabilities and investigated them with respect to MF and age, and the results are displayed graphically and narratively. A logistic regression model with sex, age at diagnosis, and EMVI was also considered; nevertheless, it was not possible to fit the model because of a problem related to complete separation [35], where EMVI was present in all of the patients who died. To show the association between EMVI and outcome, we used the probability of survival from the final multivariable model that we fitted (with sex, age, and MF) and investigated its relationship with EMVI quantitatively and graphically. All analyses were performed using the R statistical software (version 4.3.3) [36].

## 3. Results

### 3.1. Demographics

The data consisted of 44 individuals, of whom 15 (34.09%) were dead and 29 (65.91%) were alive at the time of follow-up, with only one death due to other causes. Out of the remaining 29 who were alive, 22 (75.86%) were alive with disease, and 7 (24.14%) were alive with no evidence of disease (Table 1). The one individual who died due to other causes was removed from the subsequent analysis. The results, including the descriptive statistics provided in Table 1, are from an analysis based on the 43 individuals. The median age at follow-up for the study population was 63.70 years (IQR: [58.80, 74.14]). The results indicate that those who died were relatively older (median age at follow-up = 73.18 years, IQR: [63.57, 84.64]) than those who were alive (median age at follow-up = 61.73 years, IQR: [57.63, 69.23]). The results also show that age at diagnosis was higher for those who died (median age at diagnosis = 70.91 years, IQR: [56.06, 75.94]) than those who are alive (median age at diagnosis = 54.72 years, IQR: [45.64, 61.20]), indicating that those who died were diagnosed later in their life (Table 1). The average follow-up period (from diagnosis) was 7.23 years (SD = 5.16), with the minimum follow-up of 0.11 years and a maximum of 20.99 years (Table 1). Among those who died of disease, the shortest follow-up was 0.79 years, and the longest was 12.80 years (Table 1).

### 3.2. Mesenteric Fibrogenesis and Extramural Vascular Invasion

From a histologic perspective, a large proportion of the JI NETs extended beyond the muscle wall (36/42) (Figure 4a,b).

From this group of stage pT3 and pT4 tumors, a significant percentage with available data (17/41; 41.46%) showed MF characterized by extensive areas of a fibroblastic reaction in the mesentery (Figure 5a,b).

The tumor spread through the bowel wall appeared to be accentuated at the exit points of blood vessels. In cases with MF, the fibrosis encased and tethered the vessels, predisposing to invasion, particularly in the thin-walled veins (Figure 6a,b). Indeed, our results show a high level of agreement (92.68%) between MF and EMVI, with a kappa value of 0.85 (95%CI: [0.54, 1.00]). The overwhelming majority of these MF patients died (76.47%, 13 out of 17), indicating its prognostic value (*p*-value 0.0002) (Table 2). Similarly, most patients with EMVI died (Table 1). It was also quite notable that, in those cases with EMVI, the tumor cell clusters within and around veins did not show higher proliferative activity or hotspots relative to the main tumor mass (Figure 6a,b).

A multivariable logistic regression model consisting of sex, age at diagnosis, MF, and tumor size (Table 3) showed that age at diagnosis and MF were significant predictors of survival (Figure 7a).

Considering that EMVI is highly associated with MF, we can deduce that EMVI also accurately predicts survival, with the presence of EMVI leading to reduced odds of survival. A separate model with EMVI included could not be fitted because of a complete separation problem, since EMVI was present in all of the patients who died. To indirectly examine the association between EMVI and outcomes, we explored survival probability for individuals with EMVI and without EMVI. The results (Figure 7b) show that patients without EMVI, in general, have a higher probability of survival and that EMVI behaves the same way as MF in its relationship with survival probability and age at diagnosis. Finally, we explored the relationship between the probability of survival and time from diagnosis. The results show that there is no obvious relationship between survival probability and time from diagnosis. Nevertheless, the results indeed confirm that age at diagnosis is more important than time from diagnosis in terms of predicting survival for patients with no MF.

### 3.3. Tumor Infiltration Pattern

In cases where the tumor did not involve the mesentery, it was more problematic to assess the invasive front/leading tumor edge. Unlike the pancreas, the infiltrative pattern in the luminal GI tract is different due to the differential interface provided by the various layers of the bowel wall (mucosa, submucosa, muscle wall, and serosa). This was particularly so in the muscularis propria, where the tumor tended to infiltrate in small groups in a very irregular fashion (Figure 8a), resulting in high scores; 29 tumors (69%) had a grade 3 infiltrative pattern.

For comparison, previous cases of appendiceal NETs were retrieved from our pathology archives and similarly showed grade 3 patterns; they were cured through simple appendectomy with excellent prognosis (Figure 8b). The infiltration pattern of the liver metastases was assessed in 37 specimens. It was relatively easy to do so in lobectomy and segmentectomy cases (Figure 1a). However, evaluation of some wedge resections and all core biopsies was suboptimal due to the small amount of tissue available. Overall, there is no clear link between primary or liver infiltration and survival (Table 2).

### 3.4. Tumor Grade

Most of the primary tumors (25/36) were grade 1, with only 11 that were grade 2 (Table 4). None were grade 3 in this cohort. As can be seen in Table 2, JI tumor grade was not prognostic (*p* = 0.1260). The tumor grades for the liver metastases (Figure 9a,b) were higher, with one case even attaining grade 3 (Table 4), but likewise, these grades were not prognostic (*p* = 0.2566). Nevertheless, a significant difference in tumor grade between the primaries and the liver metastases was observed (*p*-value = 0.0003).

### 3.5. Lymphovascular Invasion and Perineural Invasion

Lymphovascular invasion was observed in all the patients, and, hence, the variable is non-informative (variance = 0) as a predictor. Similarly, perineural invasion was seen in most cases (82.93%) and did not show a relationship to survival.

## 4. Discussion

Most deaths attributed to GI NETs are due to liver metastases, a preponderance of which arise from the JI [1,2,4]. This highlights an important concept for these tumors, i.e., site specificity, whereby NETs from different sites along the GI tract vary widely in their clinical and pathologic manifestations [16]. For example, appendiceal NETs, which are topographically juxtaposed to the JI in the midgut, rarely metastasize to the liver [1,2,37,38]. This accounts for their generally excellent outcomes following surgical resection. Likewise, NETs from other GI sites have their own attributes, e.g., the stomach has three distinct subtypes of NETs [39], duodenal NETs have a strong association with familial syndromes and/or gastrin/somatostatin secretion [40,41], and colonic and rectal NETs tend to be managed quite differently [42]. All of this speaks to the heterogeneity within the group. With specific reference to the JI site, the NENs therein have distinctive features, namely, (i) the neoplasms are mostly NETs, whereas NECs are rare, with only a handful of cases having been documented [16]; (ii) they are strongly associated with serotonin production and MF and (iii) have a high propensity to metastasize to the liver.

In this study, we paired JI NETs and their hepatic metastases in an effort to better understand tumor progression and prognostic determinants. Our findings show that most JI NETs were grade 1, with only a small proportion of grade 2 and none in grade 3, which is generally in accord with other studies [16,26,27,43,44]. Although there was a statistically significant difference in tumor grade between the JI NETs and the liver metastases, this did not translate into poorer outcomes. This is in line with reports in the literature [26,27,43,44]. These relatively good outcomes in the face of metastatic disease increase the management options for the patients. Hence, resection of the primary can be carried out even with stage 4 disease [45,46]. Over and above removing the source of the metastases, an additional benefit of resecting the primary is that it potentially prevents or curtails MF-induced effects, which include but are not limited to the following: (i) tethering of the adjacent bowel loops, leading to intestinal obstruction and/or intussusception, (ii) impairment of the blood supply, causing abdominal pain, aggravated diarrhea, ascites, malabsorption, and malnutrition, and, (iii) in more advanced cases, extension to the retroperitoneum, potentially causing such complications as lymphatic obstruction/chyloabdomen, obstructive uropathy, hydronephrosis, and renal failure [47].

In much the same way as with primary resection, hepatic metastasectomy, whether carried out synchronously or metachronously, can significantly improve overall survival or, in some cases, achieve cure [48]. In instances where curative intent is not possible, even cytoreductive hepatectomy can be effective in ameliorating the debilitating hormonal effects of these tumors, such as carcinoid syndrome [49]. There are also reports of liver transplantation for patients with hepatic metastases, provided that they meet certain benchmarks, e.g., Milan criteria [50,51]. The relatively indolent disease biology of JINETs described herein is in sharp contrast to other tumors with a more aggressive clinical course, such as pancreatic ductal adenocarcinoma, where the role of surgical resection in stage 4 disease is very limited and guarded, even with oligometastatic disease [52,53].

EMVI is increasingly being recognized as an important biological parameter in colorectal adenocarcinoma [54,55]. Recently, it was also shown to be prognostic for JI NETs, though there could well be different mechanisms for the development of EMVI in various neoplasms [28]. With EMVI (Figure 5 and Figure 6), the clusters of tumor cells are conveyed via the superior mesenteric vein to the portal vasculature/sinusoids. Indeed, EMVI has been likened to a ‘vascular highway’ for tumor cells, where the deep capacious venous channels allow for extensive and rapid tumor spread [28,56]. Within the clusters, the malignant cells are bound to each other and to mesenchymal cells in the accompanying tumor-associated stroma/extracellular matrix (ECM) (Figure 6a,b). It is envisaged that seeding is enhanced by joint tumor–stromal cell action. This could be a possible mechanism for explaining the heterogeneity seen in metastases without necessarily having increased tumor cell proliferation. In fact, the tumor cell clusters that invaded blood vessels did not stand out as Ki67 hotspots or mitotically hyperactive.

The tumor microenvironment (TME) is a complex nebula consisting of the extracellular matrix, stromal, endothelial, and inflammatory cells [57,58]. The interaction between tumor cells and the TME is mediated by various factors that promote tumor growth and influence tumor behavior. For JI NETs, a characteristic feature is serotonin elaboration, and it is thought to play an important role in MF [55]. However, whereas most JI NETs are positive for serotonin, MF is seen in only up to 50% of cases, which suggests that additional factors are required in the TME [57]. Implicated in this is the overexpression of genes such as COMP and COL11A1, as well as various profibrotic growth factors, including transforming growth factor beta, fibroblast growth factor, and platelet-derived growth factor [59,60]. It is also possible that a decreased expression of serotonin-metabolizing enzymes may play a part [55]. In any case, the resultant fibrogenesis encases blood vessels in the TME, predisposing them to invasion.

## 5. Limitations of the Study

Our study is subject to certain limitations, such as the relatively small sample size and retrospective design. However, since JI NETs are rare, most such prognostic studies are also tempered by these limitations. Hence, additional studies are required to further validate the generalizability of our findings.

## 6. Conclusions

This study showed that JI NETs tend to have low cell proliferation rates as measured by mitoses/Ki67, and this was also reflected in the liver metastases. As such, these parameters were not prognostic. Likewise, the concept of advancing edge profile/infiltrative pattern does not seem to work well for JI NETs on account of the differential interface presented by the histologic layers of the bowel wall. However, our study suggests that MF is an important mechanism in the metastatic cascade. MF encases and tethers mesenteric vessels, which appears to enhance EMVI, thereby conveying clusters of tumor cells to the hepatic circulation. This supports the polyclonal nature of tumor progression rather than origin from hotspot aberrant clones.

## Figures and Tables

**Figure 1 cancers-17-01486-f001:**
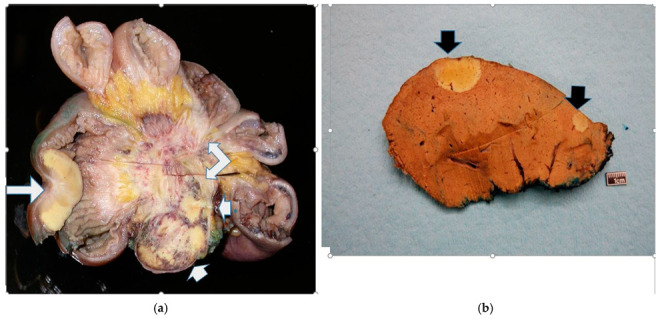
(**a**): Transverse section of a jejunoileum showing a characteristically yellow neuroendocrine tumor on the lower left side (long white arrow). The NET has spread into the mesentery (two short white arrows), causing extensive central ‘spoke-wheel type’ fibrosis (right-angled arrow) and distortion of small bowel loops (at the periphery). (**b**): Coronal section through a right hepatic lobectomy specimen showing two yellow, discrete, well-demarcated NET nodules (black arrows).

**Figure 2 cancers-17-01486-f002:**
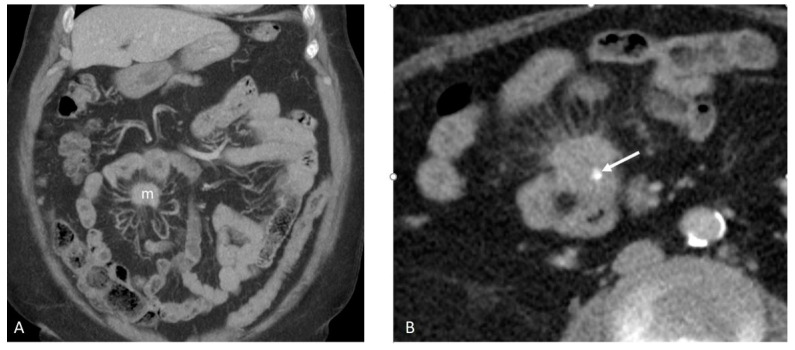
(**A**,**B**): Coronal maximum intensity project (MIP) CT image (**A**) demonstrating the mesenteric metastasis (m) in relation to the small bowel and other visceral organs. There is an abnormal configuration of the surrounding small bowel with tethering and mesenteric spiculations radiating centrally toward the metastasis without evidence of small bowel obstruction. Axial CT image of the metastasis demonstrating an internal calcification (arrow, **B**), which is commonly seen in NET metastases.

**Figure 3 cancers-17-01486-f003:**
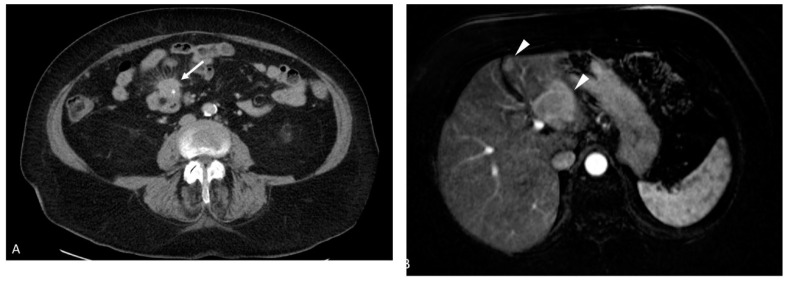
(**A**,**B**): Metastatic small bowel NET with mesenteric metastases (arrow, **A**) seen in the initial CT axial portal venous phase images and hypervascular hepatic metastases in segments 3/4 (arrowheads; **B**) demonstration of an late arterial phase axial MRI image performed 5 weeks later.

**Figure 4 cancers-17-01486-f004:**
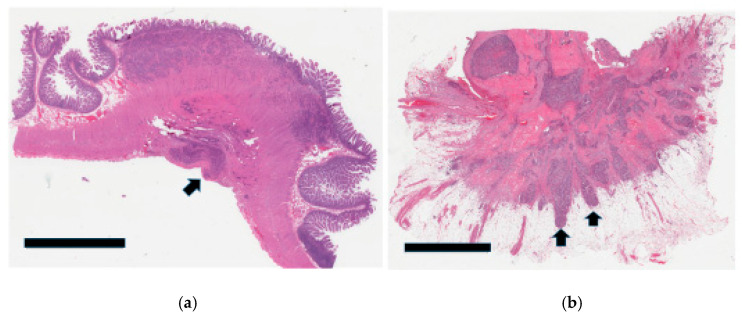
(**a**): Jejunoileal NET showing transmural involvement of the bowel wall with focal extension to reach the serosal surface (arrow). (Black bar scale at bottom left = 5 mm). (**b**): Mesenteric involvement by a JI NET depicting tumor nodules in a markedly fibroblastic stroma. There is EMVI extending as tumor nubbins (arrows) on the deep aspect. (Black bar scale at bottom left = 7 mm).

**Figure 5 cancers-17-01486-f005:**
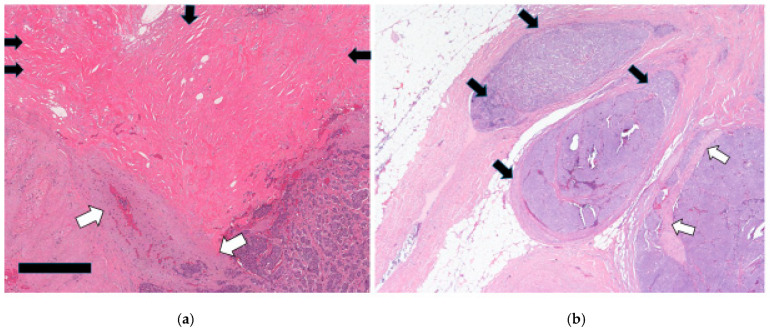
(**a**): Tumor in the lower-right corner with a severe fibroblastic reaction in the upper field (black arrows). It has encased a vessel (white arrows) adjacent to the left side of the tumor with resultant EMVI. (Black bar scale at bottom left = 0.4 mm). (**b**): A micrograph showing tumor nodules (black arrows) within the lumen of these fairly large mesenteric vessels. There is also perineural invasion (white arrows) in the nodule on the bottom-right corner.

**Figure 6 cancers-17-01486-f006:**
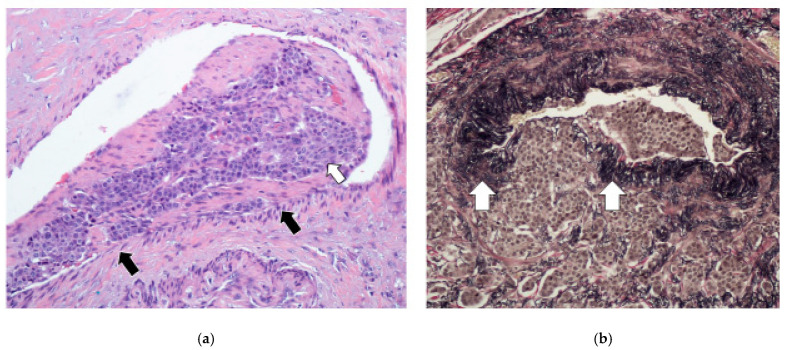
(**a**): A high magnification of the tumor cells (white arrow) in the vein (black arrows—vein wall) to show the organoid morphology typical of well-differentiated NETs. The cells are generally similar, with mostly round nuclei and hardly any mitotic activity, which is consistent with grade 1. (**b**): A van Gieson stain (of the same area as (**a**)) showing disruption of the elastic lamina (white arrows) (lower-left side of the vein) by the tumor that is now growing in the lumen. Note the organoid and relatively bland morphology of the invading cells.

**Figure 7 cancers-17-01486-f007:**
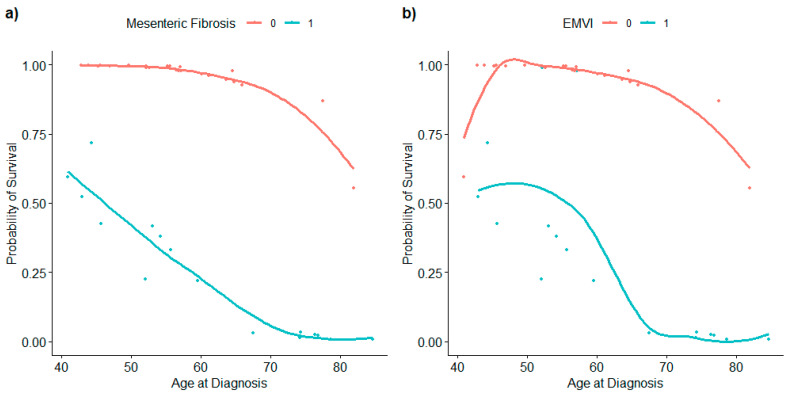
(**a**) MF is a very important variable for predicting the survival of patients with WD JINETs. However, age is also important since it is the deriving factor in predicting survival for patients with no MF, where survival probability declines monotonically with respect to age. (**b**) Exploration of the survival probability for individuals with EMVI and without EMVI. Patients without EMVI generally have a higher probability of survival. EMVI behaves the same way as MF in its relationship with survival probability and age at diagnosis.

**Figure 8 cancers-17-01486-f008:**
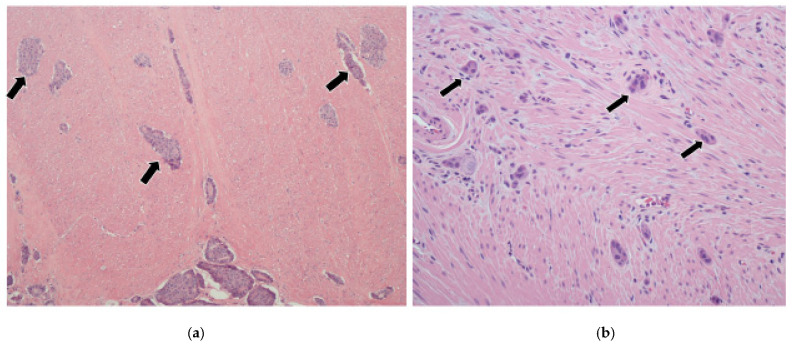
(**a**) Section from a JI NET showing the advancing edge of the tumor within the muscle wall. The tumor cell groups (black arrows) tend to be dispersed in a very irregular infiltrative pattern as they traverse through this interface. (**b**) For comparison, a well-differentiated appendiceal NET shows a similar phenomenon (black arrows = tumor nodules) without necessarily connoting biologic aggressiveness.

**Figure 9 cancers-17-01486-f009:**
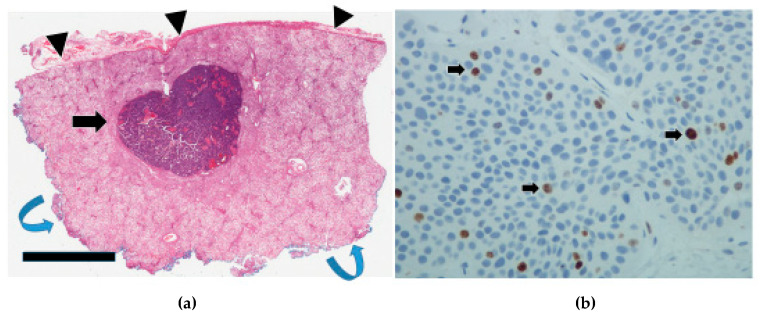
(**a**): Section of a wedge resection of a liver showing a well-defined metastatic nodule (black arrow). Glisson’s capsule (black arrowheads) is at the top. The surgical excision margin has been inked in blue (blue curved arrows) for orientation purposes. (Black bar scale at bottom left = 6 mm). (**b**): This Ki67 stain of the liver shows Ki67-positive cells (black arrows) consistent with a grade 2 NET.

**Table 1 cancers-17-01486-t001:** Summary statistics characterizing patients with well-differentiated jejunoileal neuroendocrine tumors.

Variables	Frequency (Percentage) *		
	Overall (N = 43)	Alive (N = 29)	Dead (N = 14)
Age at follow-up			
Median (IQR)	63.70 (58.80, 74.14)	61.73 (57.63, 69.23)	73.18 (63.57, 84.64)
Mean (SD)	66.32 (12.18)	63.30 (9.80)	72.57 (14.49)
(Min, Max)	(43.70, 97.53)	(43.70, 83.29)	(46.45, 97.53)
Age at diagnosis			
Median (IQR)	56.53 (50.93, 66.77)	54.72 (45.64, 61.20)	70.91 (56.06, 75.94)
Mean (SD)	59.08 (12.21)	55.48 (10.68)	66.55 (12.14)
(Min, Max)	(40.91, 84.73)	(40.91, 81.93)	(45.66, 84.73)
Follow-up time			
Median (IQR)	5.81 (4.42, 9.44)	6.32 (4.61, 9.87)	6.02 (3.68)
Mean (SD)	7.23 (5.16)	7.82 (5.70)	5.59 (3.01, 8.74)
(Min, Max)	(0.11, 20.99)	(0.11, 20.99)	(0.79, 12.80)
Sex			
Female	18 (41.86)	10 (34.48)	8 (57.14)
Male	25 (58.14)	19 (65.52)	6 (942.86)
Tumor Stage **			
1	2 (4.76)	0 (0)	2 (14.29)
2	4 (9.52)	3 (10.71)	1 (7.14)
3	10 (23.81)	7 (25.00)	3 (21.43)
4	26 (61.90)	18 (64.29)	8 (57.14)
Tumor Size: mean (SD)	1.82 (0.70)	1.72 (0.50)	2.02 (0.99)
Lymph node stage **			
0	0 (0)	0 (0)	0 (0)
1	25 (59.52)	17 (60.71)	8 (57.14)
2	17 (40.48)	11 (39.29)	6 (42.86)
Mesenteric fibrosis **			
No	24 (58.54)	23 (82.14)	1 (7.14)
Yes	17 (41.46)	4 (14.29)	13 (92.86)
EMVI **			
No	23 (56.10)	23 (85.19)	0 (0)
Yes	18 (43.90)	4 (14.81)	14 (100)
Perineural invasion **			
No	6 (14.63)	2 (7.41)	4 (30.77)
Yes	34 (82.93)	25 (92.59)	9 (69.23)

* Frequency and percentages are given unless it is specified otherwise. ** The denominator for calculating percentages does not include missing data.

**Table 2 cancers-17-01486-t002:** One-variable (unadjusted) logistic regression models where the binary dependent (outcome) variable, survival (dead = 0, alive = 1), is fitted against each potential predictor.

Potential Predictor Variables	Unadjusted OR [95% CI]	*p*-Value
Sex (Female)	0.39 [0.10, 1.44]	0.1632
Age at diagnosis (in years)	0.92 [0.86, 0.98]	0.0089 *
Age at follow-up (in years)	0.93 [0.87, 0.99]	0.0262 *
Presence of mesenteric fibrosis	0.013 [0.001, 0.093]	0.0002 *
Tumor stage	1.53 [0.73, 3.35]	0.2580
Tumor size	0.54 [0.20, 1.36]	0.1949
Lymph node stage	0.86 [0.23, 3.26]	0.8240
Perineural invasion	5.56 [0.92, 45.39]	0.0708
Primary Ki67	5.62 [0.86, 111.89]	0.1260
Liver mitosis	0.76 [0.16, 4.26]	0.7410
Liver Ki67	0.47 [0.11, 1.67]	0.2566

* Statistically significant at the α = 0.05 level of significance.

**Table 3 cancers-17-01486-t003:** Results from the multivariable logistic regression model where the binary dependent (outcome) variable is survival (dead = 0, alive = 1).

Potential Predictor Variables	Adjusted OR [95% CI]	*p*-Value
Sex (Female)	3.64 [0.25, 119.84]	0.3972
Age at diagnosis	0.86 [0.70, 0.96]	0.0441 *
Mesenteric fibrosis	0.002 [0.000, 0.059]	0.0091 *
Tumor Size	1.50 [0.19, 13.18]	0.6901

* Statistically significant at the α = 0.05 level of significance.

**Table 4 cancers-17-01486-t004:** Summary data on the tumor grade and infiltrative pattern of primary and liver metastases for patients with well-differentiated jejunoileal neuroendocrine tumors.

Markers	Frequency (Percentage)		
	Overall	Alive	Dead
Primary infiltration			
1	6 (14.29)	6 (21.43)	0 (0)
2	7 (16.67)	7 (25.00)	0 (0)
3	29 (69.05)	15 (53.57)	14 (100)
Primary mitosis			
1	38 (95.00)	26 (92.86)	12 (100)
2	2 (5.00)	2 (7.14)	0 (0)
3	0 (0)	0 (0)	0 (0)
Primary Ki67			
1	25 (69.44)	16 (61.54)	9 (90.00)
2	11 (30.56)	10 (38.46)	1 (10.00)
3	0 (0)	0 (0)	0 (0)
Liver infiltration			
1	22 (59.46)	22 (84.62)	0 (0)
2	12 (32.43)	4 (15.38)	8 (72.73)
3	3 (8.11)	0 (0)	3 (27.27)
Liver mitosis			
1	35 (81.40)	24 (82.76)	11 (78.57)
2	8 (18.60)	5 (17.24)	3 (21.43)
3	0 (0)	0 (0)	0 (0)
Liver Ki67			
1	17 (42.50)	13 (46.43)	4 (33.33)
2	22 (55.00)	15 (53.57)	7 (58.33)
3	1 (2.50)	0 (0)	1 (8.33)

## Data Availability

The raw data supporting the conclusions of this article will be made available by the authors on request.

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
