# Peer review of "Well-Differentiated Jejunoileal Neuroendocrine Tumors and Corresponding Liver Metastases: Mesenteric Fibrogenesis and Extramural Vascular Invasion in Tumor Progression"

_cancers, 2025, doi:10.3390/cancers17091486_

Round 1
Reviewer 1 Report
Comments and Suggestions for Authors
THe topic is of interest to the readership. However, there are these limitations:
1) The sample size is limited and the design is retrospective, so prone to selection bias
2) The series seems quite heterogeneous
3) The authors should test the normality of the distribution through the Kolmogorov-Smirnov test
4) The ORs were adjusted for which variables?
5) The discussion should be improved. The authors should provide some comments on the potential benefit of resection of the primary tumor in well differentiated NETs with liver metastases (cite the series PMID: 27956320)
Author Response
Reviewer #1
The topic is of interest to the readership. However, there are these limitations:
1) The sample size is limited and the design is retrospective, so prone to selection bias. We completely agree. However, these are rare tumors which also makes it difficult to study them prospectively. Therefore, we have now added a paragraph (after the Discussion) entitled 'Limitations of the study'.
2) The series seems quite heterogeneous. This is true. The common denominator in the cohort is well-differentiated JI NETs with liver metastases.
3) The authors should test the normality of the distribution through the Kolmogorov-Smirnov test. Our outcome is binary (alive=1, dead=0), hence the distribution is Binomial and not normal. Therefore, we used the logistic regression model, which is a generalized linear model (GLM) with a binomial distribution (as appropriate for binary outcome) and the logit link function. We did not do any statistical inference involving a continuous variable that requires checking for the normality assumption. We have re-worded the Statistics selection (with additional references) to provide better clarity (please see marked up copy).
4) The ORs were adjusted for which variables? The final model consists of sex, age at diagnosis, MF and tumor size. These variables with the corresponding adjusted odds ratios are listed in Table 3. Since the model involves more than one variable, the odds ratio corresponding to one variable that we get from the logistic regression is adjusted for the other variables in the model, hence the term adjusted odds ratio (we have also added a reference).
5) The discussion should be improved. The authors should provide some comments on the potential benefit of resection of the primary tumor in well differentiated NETs with liver metastases (cite the series PMID: 27956320). We have improved the discussion and put in additional references including PMID: 27956320. Please see 'marked up' copy.
Reviewer 2 Report
Comments and Suggestions for Authors
The manuscript by Ranot JM and co-authors presented an observation of the association between two pathological markers and prognosis prediction for the jejunoileal neuroendocrine tumors. The authors show that mesenteric fibrogenesis and extramural vascular invasion in tumor progression had the highest significance correlation with poor prognosis in this type of tumor. There are indications that regular proliferation marker KI67 or a grade of the tumor, which usually indicates high speed progression of the tumor, is not good in this case. The papers should be interesting for the readers of the Cancers journal, but some comments need to be addressed.
1) The paper is too observational without trying to prove the concept experimentally
2) Figures 5 - 9 showing histologic photos should be supplemented with arrows for the convenience of the readers
3) NETs abbreviation next to neuroendocrine tumors recently used for the indications of NETosis
4) The discussion is too short. Mesenteric fibrogenesis and extramural vascular invasion should be described in the Introduction. Mesenteric fibrogenesis as a tumor-induced state should also be discussed. How it affects the tumor microenvironment?
Author Response
Reviewer #2
The papers should be interesting for the readers of the Cancers journal, but some comments need to be addressed.
1) The paper is too observational without trying to prove the concept experimentally. This is true. However, as pointed out by PMID: 27956320, prognostic investigations on NETs are limited by the rare/uncommon nature of these tumors and the retrospective design of the studies. We have now added a paragraph to describe the limitations of the study.
2) Figures 5 - 9 showing histologic photos should be supplemented with arrows for the convenience of the readers. More arrows have been added to highlight key features.
3) NETs abbreviation next to neuroendocrine tumors recently used for the indications of NETosis. Yes, we do see this. It is unfortunate since the usage of NET for neuroendocrine tumors predates 'NETosis' (e.g. ENETS, NANETS). Hopefully, the context will help to clarify the meaning.
4) The discussion is too short. Mesenteric fibrogenesis and extramural vascular invasion should be described in the Introduction. Mesenteric fibrogenesis as a tumor-induced state should also be discussed. How it affects the tumor microenvironment? The manuscript has been revised to cover all these points and is now considerably longer (please see the 'marked up' copy).
Reviewer 3 Report
Comments and Suggestions for Authors
The manuscript by Ranot and co-authors describes an investigation of the characteristics and behavior of jejunoileal neuroendocrine tumors (JI NETs), particularly focusing on their cell proliferation rates and metastatic mechanisms. The findings of the authors that JI NETs exhibit low cell proliferation rates, as measured by mitoses/Ki67, and that these rates are consistent in liver metastases, are well supported by the data presented. The conclusion that these parameters are not prognostic is significant, in view of the existing assumptions about the predictive value of cell proliferation in these tumors. The results are important and deserve publication.
Specific comments:
Figure 4: The inscriptions denoting nanometers are hardly visible in the pictures. The same applies to Figures 5 and 9.
Figure 7: The resolution of the figure is low and should be enhanced.
A more detailed discussion of the clinical implications of the findings in Discussion section is desirable.
Summarizing, I recommend acceptance of the manuscript for publication after minor revision.
Author Response
Reviewer #3
Figure 4: The inscriptions denoting nanometers are hardly visible in the pictures. The same applies to Figures 5 and 9. We have now corrected this.
Figure 7: The resolution of the figure is low and should be enhanced. We have enhanced the resolution.
A more detailed discussion of the clinical implications of the findings in Discussion section is desirable. This has now been done (please see 'marked up' copy).
Summarizing, I recommend acceptance of the manuscript for publication after minor revision. We have now revised the manuscript in line with the recommendations. Thank you.
Round 2
Reviewer 1 Report
Comments and Suggestions for Authors
The manuscript is OK